# Genome sequencing and functional characterization of a *Dictyopanus pusillus* fungal enzymatic extract offers a promising alternative for lignocellulose pretreatment of oil palm residues

Andrés M. Rueda[1,2,3], Yossef López de los Santos[1], Antony T. Vincent[1], Myriam Létourneau[1], Inés Hernández[3], Clara I. Sánchez[3,4], Daniel Molina V.[5], Sonia A. Ospina[2], Frédéric J. Veyrier[1], Nicolas Doucet[1,6] *

1 Centre Armand-Frappier Santé Biotechnologie, Institut National de la Recherche Scientifique (INRS), Université du Québec, Laval, Canada, 2 Instituto de Biotecnología, Universidad Nacional de Colombia, Bogotá, Colombia, 3 Centro de Estudios e Investigaciones Ambientales, Universidad Industrial de Santander, Bucaramanga, Colombia, 4 Escuela de Microbiología, Universidad Industrial de Santander, Bucaramanga, Colombia, 5 Escuela de Química, Universidad Industrial de Santander, Bucaramanga, Colombia, 6 PROTEO, Québec Network for Research on Protein Function, Engineering, and Applications, Québec, Canada

* nicolas.doucet@inrs.ca

## Abstract

The pretreatment of biomass remains a critical requirement for bio-renewable fuel production from lignocellulose. Although current processes primarily involve chemical and physical approaches, the biological breakdown of lignin using enzymes and microorganisms is quickly becoming an interesting eco-friendly alternative to classical processes. As a result, bioprospection of wild fungi from naturally occurring lignin-rich sources remains a suitable method to uncover and isolate new species exhibiting ligninolytic activity. In this study, wild species of white rot fungi were collected from Colombian forests based on their natural wood decay ability and high capacity to secrete oxidoreductases with high affinity for phenolic polymers such as lignin. Based on high activity obtained from solid-state fermentation using a lignocellulose source from oil palm as matrix, we describe the isolation and whole-genome sequencing of *Dictyopanus pusillus*, a wild basidiomycete fungus exhibiting ABTS oxidation as an indication of laccase activity. Functional characterization of a crude enzymatic extract identified laccase activity as the main enzymatic contributor to fungal extracts, an observation supported by the identification of 13 putative genes encoding for homologous laccases in the genome. To the best of our knowledge, this represents the first report of an enzymatic extract exhibiting laccase activity in the *Dictyopanus* genera, offering means to exploit this species and its enzymes for the delignification process of lignocellulosic by-products from oil palm.

**Data Availability Statement:** The Whole Genome Shotgun project was deposited at DDBJ/ENA/ GenBank under the accession QVIE00000000.

**Funding:** This work was partially supported by a grant from Universidad Industrial de Santander, Vicerrectoria de Investigación y Extension (Grant number 5199) (to C.I.S. and D.M.V.), Industrias Acuña INAL LTDA (Grant number 8712) (to C.I.S.), and a Natural Sciences and Engineering Research Council of Canada (NSERC), via Discovery Grant RGPIN-2016-05557 (to N.D.). A.T.V. received a Postdoctoral Fellowship from NSERC. A.M.R. was supported by a doctoral scholarship from the Colombian Departamento Administrativo de Ciencia, Tecnología e Innovación (Colciencias) (PhD scholarship 567, 2012), and was the recipient of a scholarship from the Emerging Leaders in the Americas Program (ELAP) from the Government of Canada. F.V. and N.D. hold Fonds de Recherche Québec-Santé (FRQS) Research Scholar Junior 1 and Senior Career Awards, respectively (numbers 35038 and 281993). There was no additional external funding received for this study.

**Competing interests:** Private funding provided by Industrias Acuña INAL LTDA (Grant number 8712) resulted from an internal competition by the Universidad Industrial de Santander (UIS) to support part of the current research project performed in the laboratory of Clara I. Sánchez at UIS (Colombia). Industrias Acuña INAL LTDA does not commercially benefit from the publication of this manuscript, nor was it involved in the conception, design, processing, analysis, or communication of the results and research project. All authors are fully committed to the full disclosure of all results and research materials resulting from the current research. All authors proclaim that this declaration of competing interests does not alter their adherence to all the PLOS ONE policies on sharing data and material.

# 1. Introduction

The accumulation of agro-industry lignocellulosic postharvest by-products is a direct consequence of the global demand for crops employed in the food supply chain and bio-renewable fuel production. Following this trend, global palm oil production has increased 41% over the past 10 years to reach 71.45 million tons in 2018, primarily due to high biodiesel demand [1]. As a result, the product-to-waste ratio for palm oil production remains significantly high (1:3), generating important lignocellulosic biomass accumulation [2]. This represents a particularly pressing environmental issue for the largest producing countries such as Malaysia and Indonesia. One alternative to overcome the significant build-up of cellulosic biomass is the production of bioethanol by fermentation of syrups extracted from cellulose and hemicellulose hydrolysis. Lignocellulosic ethanol production is an eco-friendly alternative to current agro-industry by-products, in addition to offering an important source of renewable energy [3].

Lignocellulose is a raw material composed of lignin, cellulose, and hemicellulose, forming a complex aromatic polymer that provides rigidity and strength to plant cell walls. While cellulose represents an inestimable carbon energy source on a global scale, releasing cellulose from lignocellulose by lignin removal represents a major challenge in many industrial processes, including the bioethanol and pulp and paper industries [4–6]. To this day, delignification is either performed by chemical strategies using environmentally damaging acids or alkaline solutions, and/or through physical processes such as high temperature and pressure conditions [7]. A biological delignification process using ligninolytic enzymes that breakdown lignin through an oxidation mechanism would therefore offer a valuable alternative for the pretreatment of lignocellulose [8]. Laccases (EC 1.10.3.2), manganese peroxidases (EC 1.11.1.13), and lignin peroxidases (EC 1.11.1.14) are the most promising ligninolytic catalysts for such biological pretreatment. These enzymes are primarily expressed and secreted from basidiomycete fungi, especially the *Agaricomycetes* class [9]. Fungi are the main organisms associated to wood decay colonization due to their ability to secrete oxidoreductases and their high affinity for phenolic polymers such as lignin. Studies on fungi lignocellulose decomposition have thus demonstrated that species involved in wood decay produce a pool of many enzymes acting against the three primary lignocellulose components [10,11].

It has been established that co-evolution between white-rot fungi and angiosperms favored the specialization of ligninolytic enzymes to degrade lignin and a broad range of compounds derived from wood decay, turning these organisms into valuable biotechnological tools [12,13]. Fungal enzymatic extracts exhibiting ligninolytic activities are thus currently positioned as a promising biotechnological tool for the management of recalcitrant pollutants such as dyes, pesticides, phenolic compounds, and agro-industry residues [14,15]. Nevertheless, fungus-based lignocellulosic pretreatment processes for industrial applications is still hampered by the difficulty to produce large amounts of highly active enzymes. Luckily, these problems can partly be overcome by the use of recombinant organisms and/or screening of species with enhanced enzymatic ability [16,17]. Additionally, new sequencing techniques used in combination with fungi bioprospecting can increase our understanding of the enzymatic delignification process performed by fungi during lignocellulose recycling. Such knowledge can then serve as basis to develop biotechnological alternatives to handle lignocellulosic residues from agro-industry, potentially leading to new developments in the production of bioethanol and/or organic compounds [18–20].

Herein, we describe the isolation, whole-genome sequencing of *D. pusillus*, and initial characterization of wild basidiomycete enzymatic extracts exhibiting ABTS oxidation as an indicative of laccase activity. To shed light on potential enzymes involved in this ligninolytic activity, the genome of *D. pusillus* was sequenced and annotated using single-molecule real-time

sequencing technology. Our overall strategy for bioprospecting, fungi isolation & identification, experimental characterization of ligninolytic activity and genome sequencing is summarized in S1 Fig in S1 File. Our main goal was to identify new fungal enzymatic tools capable of sustaining harsh experimental conditions for extended periods of time, such as higher temperatures and lower pH, while favoring an increase in the release of reducing sugars during simultaneous pretreatment and saccharification processes of empty fruit bunch from oil palm trees. We found that laccase activity was the main enzymatic contributor to our fungal enzymatic extracts, which included a highly active isolate from *D. pusillus* LMB4. In addition to characterizing potentially valuable biotechnological tools for the enzymatic lignocellulose pretreatment of oil palm tree residues, our results also present the first complete genome sequencing of a *Dictyopanus* fungus.

## 2. Materials and methods

### 2.1. Fungi isolation and growth conditions

Fruit bodies from basidiomycete fungi growing on decaying wood were collected in a tropical humid forest in Colombia, following previously published parameters to favor the presence of delignification enzymes [21,22]. The main inclusion criteria were macroscopic properties belonging to the orders of *Agaricales*, *Russulales*, and *Polyporales* due to the possible ligninolytic activity of these organisms [23,24]. Collected samples were kept in wax paper bags to prevent deterioration. Isolation of the collected fungi was performed in wheat bran extract agar composed of 18 g.L$^{-1}$ agar, 10 g.L$^{-1}$ glucose, 5 g.L$^{-1}$ peptone, 2 g.L$^{-1}$ yeast extract, 0.1 g.L$^{-1}$ KH$_2$PO$_4$, 0.1 g.L$^{-1}$ MgSO$_4$.7H$_2$O, 0.085 g.L$^{-1}$ MnSO$_4$, 0.1 g.L$^{-1}$ chloramphenicol, 0.1 g.L$^{-1}$, 600 U.L$^{-1}$ nystatin, and 1000 ml wheat bran extract. Wheat bran extract was obtained by filtering 175 g.L$^{-1}$ of wheat brand soaked in distilled water for 1 h. Pilei were adhered to the top cover of Petri dishes, allowing spores to fall and, eventually, to germinate on the culture media. Top covers were rotated every 24 h for 3 days and those containing the pilei were replaced by new sterilized ones [25]. Sub-cultures in the same media were incubated at 25˚C to obtain axenic strains from these isolates. The axenic cultures were determined by fungal slide culture technique [26]. The presence of microscopic sexual basidiomycete properties was checked, including septate hyaline hyphae and clamps. Lactophenol cotton blue stain was used for all the microscopic observations. Twelve ligninolytic fungi belonging to genera *Aleurodiscus*, *Dictyopanus*, *Hyphodontia*, *Mycoacia*, *Phellinus*, *Pleurotus*, *Stereum*, *Trametes*, and *Tyromyces* were axenically isolated from 43 collected wild-type strains. Fungi collection was planned under the regulations of Colombia's Environmental Ministry. The research permit in biological biodiversity was obtained from the *Corporación Autónoma de Santander* (file number 153–12 REB) and with the agreement of the *Ministerio del Interior*, certifying the absence of ethnic groups in the area (application number 1648, August 14, 2012).

### 2.2. Phylogenetic identification of selected isolates

Total genomic DNA was extracted from selected isolates following a standard phenol-chloroform protocol. Briefly, fungi were grown in wheat bran extract agar for 15 days and 0.5 g of mycelium was placed in a tube with a lysis solution (0.1 M NaCl$_2$, Tris-HCl pH 8, 5% SDS) and 0.5 mm diameter glass beads. The aqueous fraction was collected, and the fungus DNA was precipitated with isopropanol. The DNA pellet was dissolved in TE buffer (10 mM Tris, 1 mM EDTA, pH 8.0) [27]. A pair of primers within the Internal Transcribed Spacer regions (ITS1/ITS4) was used to amplify ribosomal DNA by PCR [28]. PCR products were sequenced by the Sanger method using the same amplification primers. ITS1 sequences were used as query to retrieve the most similar DNA sequences from the NCBI database. A set of 36 curated

sequences were extracted from the results obtained through BLAST, after which the ITS1 sequences and the query sequence were used to create a multiple sequence alignment. To infer the evolutionary history and obtain the genetic identity of the fungus isolated and pre-identified as *Dictyopanus* sp., we applied the UPGMA protocol, where the best tree hits arose after a bootstrap of 500 repetitions using the Maximum Composite Likelihood method to obtain the evolutionary distances. All phylogenetic analyses were performed with the MEGA suite, version 10.0.5 [29].

## 2.3. Fiber analysis of oil palm by-products

Neutral Detergent Fiber (NDF), Acid Detergent Fiber (ADF), and Acid Detergent Lignin (ADL) were determined by the Van Soest method using the FiberCap™ system (Foss Analytical AB, Denmark). Cellulose and hemicellulose percentages were estimated as the difference between ADF and ADL, and NDF and ADF respectively, while lignin concentrations corresponded to ADL percentages in dry weight of oil palm by-products. Additionally, values were used to estimate the total carbon concentration in fermentation assays. All assays were performed in duplicate.

## 2.4. Basidiomycete screening by Solid-State Fermentation (SSF)

The main selection criterion of isolated wild-type fungi was ligninolytic activity observed in the crude fungi enzymatic extracts from SSF using lignocellulosic material from oil palm by-products [30]. SSF was performed in 250 ml flasks in sterile conditions. Each flask contained 12 ml of basal media in deionized water, comprising 0.2 $g.L^{-1}$ yeast extract, 0.76 $g.L^{-1}$ peptone, 0.3 $g.L^{-1}$ urea, 0.25 $g.L^{-1}$ $CuSO_4 \cdot 5H_2O$, 1.4 $g.L^{-1}$ $(NH_4)_2SO_4$, 2 $g.L^{-1}$ $KH_2PO_4$, 0.3 $g.L^{-1}$ $MgSO_4 \cdot 7H_2O$, 0.4 $g.L^{-1}$ $CaCl_2 \cdot 2H_2O$, 0.005 $g.L^{-1}$ $FeSO_4 \cdot 7H_2O$, 0.0016 $g.L^{-1}$ $MnSO_4$, 0.0037 $g.L^{-1}$ $ZnSO_4 \cdot 7H_2O$, 0.0037 $g.L^{-1}$ $CoCl_2 \cdot 6H_2O$, and 2.5 $g.L^{-1}$ of empty fruit bunch chopped into chunks of approximately 2 $cm^3$. Each flask was inoculated with eight agar plugs cut from actively growing fungal mycelium grown on wheat bran extract agar. Each SSF batch isolation contained thirty flasks and fermentation was held without agitation at 25˚C for 30 days. Every three days, three flasks were used to collect crude enzymatic extracts.

## 2.5. Recovery of crude fungal enzymatic extracts

Crude fungal enzymatic extracts were obtained by addition of 30 ml of 60 mM sterile phosphate buffer into the fermentation flask, which was shaken for 24h at 150 rpm. Whole flask contents were then collected in 50 ml tubes, vortexed in a Benchmark Scientific multi-tube vortexer for 15 minutes at 1500 rpm, and finally centrifuged twice at 8900*g* for 15 minutes to remove suspended solids. Supernatants were taken as crude fungal enzymatic extracts [31] and concentrated by lyophilization to evaluate the effects of pH and temperature on enzymatic activity and simultaneous pretreatment and saccharification.

## 2.6. Quantification of reducing sugars

Reducing sugars were quantified by oxidation of 3,5-dinitrosalicylic acid to 3-amino,5-nitrosalicylic acid (DNS) by measuring the release of the reducing extremity of sugars. The reaction was followed at 420 nm and a standard curve was obtained with glucose (0,1 to 1 $mg.ml^{-1}$) to quantify the concentration of reducing sugars [32].

## 2.7. Ligninolytic and cellulase assays

Crude fungal enzymatic extracts obtained from SSF were assayed for laccase, lignin peroxidase, and manganese peroxidase activities. Laccase activity was followed by the oxidation of 2,2′-azino-bis (3-ethylbenzothiazoline-6-sulfonic acid) (ABTS) (Sigma-Aldrich, USA) [33]. Reactions were initiated by mixing 40 μL of culture supernatant, 150 μL of 50 mM acetate buffer (pH 4.5) and 10 μL of 1.8 mM ABTS; activity of mixtures was estimated by reading absorbance at 420 nm. Manganese peroxidase activity was measured by the formation of $Mn^{3+}$-malonate complexes at pH 4.5 in 50 mM sodium malonate buffer containing 0.5 mM $MnSO_4$ [34]. Reactions were performed by mixing 20 μl of culture supernatant, 100 μl of 20 mM citrate buffer at pH 4.5, 40 μl of sodium malonate buffer, and initiated with 40 μl of fresh 0.8 mM $H_2O_2$. Readings at 270 nm were used to estimate the transformation of $Mn^{+3}$ to $Mn^{+2}$ as manganese peroxidase activity. Lignin peroxidase activity was measured by the transformation of 3,4-dimethoxybenzyl alcohol (VA) (Sigma-Aldrich, USA) to veratryl aldehyde (VAD), which exhibits a yellow color [35]. Reactions were performed by mixing 20 μl of culture supernatant, 100 μl of 20 mM citrate buffer at pH 3, 40 μl of 10 mM VA, and initiated with 40 μl of fresh 0.8 mM $H_2O_2$. Enzymatic activity was measured at 310 nm and expressed in units per liter (U.L$^{-1}$). One unit of enzymatic activity was defined as the quantity of enzyme needed to transform 1 μmol of substrate per minute. Absorbance readings were performed with a ThermoFisher Multiskan™ GO Microplate Spectrophotometer.

The total cellulosic activity was quantified by units of paper filter (UPF.ml$^{-1}$). In tubes, 500 μL of commercial cellulase solutions from *Trichoderma reesei* Sigma Aldrich C2730 Celluclast® (USA) were incubated with 500 μL of 50 mM citrate buffer at pH 4.8, 50 and 5 mg of filter paper for 1 h, at 50 ˚C. The concentration of reducing sugars released was measured by the oxidation of 3,5-dinitrosalicylic acid (DNS), as described above [36].

## 2.8. Effect of pH and temperature on ABTS oxidation as laccase activity

The effect of pH was examined for crude fungal enzymatic extracts exhibiting the highest laccase activity. A pH range from 2 to 8 (50 mM hydrochloric acid buffer, pH 2; 50 mM citric buffer pH 3–4; 50 mM acetate buffer pH 4.5–5, and 50 mM phosphate buffer pH 6–8) was evaluated using ABTS as substrate. The effect of temperature on enzyme activity and stability was measured with crude fungal enzymatic extracts in 50 mM acetate buffer pH 4.5 at 40˚C, 50˚C, and 60˚C for 7 h. Finally, comparison of crude fungal enzymatic extracts with a control laccase from *Trametes versicolor*, 53739 Sigma-Aldrich (Canada) was performed in triplicate using pH and temperature conditions exhibiting the highest activity. All components (except enzymes) were sterilized separately and mixed under environmentally sterile conditions.

## 2.9. Simultaneous pretreatment and saccharification of empty fruit bunch

The simultaneous pretreatment and saccharification process was performed in 50 ml tubes containing 1.5 g empty fruit bunch, 16 ml of 50 mM acetate buffer at pH 4.5 and combining either the laccase enzyme from *D. pusillus* or the commercial laccase from *T. versicolor* (53739 Sigma-Aldrich-Canada) with the cellulase from *T. reesei* (Sigma Aldrich C2730 Celluclast®). For the reaction mixture, both laccase and cellulase were added in a volume of 2 ml to reach a final concentration of 25 $U_*L^{-1}$ and 50 UPF, respectively. Tubes were incubated at 40˚C for 72 hours. The saccharification process was evaluated by the production of reducing sugars, measured by a DNS assay. Assays were performed in triplicate and all components (except enzymes) were sterilized separately and mixed under environmentally sterile conditions.

Simultaneous pretreatment and saccharification of empty fruit bunch was conducted with fungal enzymatic extracts exhibiting laccase activity and cellulases according to a multilevel

factorial experimental design (3 levels with 5 variables) to evaluate significant variables in the experimental process [37]. Five independent variables were evaluated: pH (3 to 5) using either 50 mM acetate buffer (pH 3 and 4) or 50 mM citrate buffer (pH 5), temperature (25, 35, and 45 ˚C), copper concentration (1, 3, and 5 mM), laccase (100, 200, and 300 U.L$^{-1}$), and cellulase (50, 100, 150 UPF.ml$^1$) activities. Simultaneous pretreatment and saccharification was performed in 50 ml tubes with 1.5 g empty fruit bunch and 20 ml total volume, including 2 ml each of laccase enzymatic extract and cellulase concentrate. The mixture was incubated for 72 h and the concentration of reducing sugars was measured in each tube. To increase robustness of the analysis, 4 experimental replicates were performed and results were analyzed with a confidence interval of 95% using Statgraphics Centurion XVII.

## 2.10. Genome analysis of *D. pusillus* LMB4

Mycelium from *D. pusillus* grown on Potato Dextrose Agar (PDA) was used to extract the genomic DNA (gDNA) through a high salt phenol-chloroform cleanup protocol recommended by PacBio® systems. More precisely, 0.5 g of mycelium was placed in a tube with a lysis solution (0.1 M NaCl$_2$, Tris-HCl pH 8, 5% SDS) and 0.5 mm diameter glass beads until mycelium was broken (visual evaluation) and centrifuged at 11,000$g$ for 10 minutes. The supernatant was mixed in the same proportion with a phenol-chloroform-isoamyl alcohol solution 25:24:1 and centrifuged at 11,000$g$ for 5 minutes. The new supernatant was mixed again in the same proportion with a chloroform-isoamyl alcohol solution (24:1) and centrifuged at 14,000$g$ for 10 minutes. Finally, the aqueous fraction was collected and fungal proteins were precipitated by adding absolute ethanol (10:3 aqueous fraction-ethanol). After centrifugation at 11,000$g$ for 15 minutes, the supernatant was mixed with ethanol (10:17 supernatant-ethanol) to precipitate DNA. The DNA pellet was obtained by centrifugation at 11,000$g$ for 15 minutes and dissolved in DEPC-treated DNase-free water. The genomic DNA of *D. pusillus* LMB4 was sequenced using five SMRT cells on a Pacific Biosciences RS II system at the Génome Québec Innovation Centre (McGill University, Montréal, Canada). The 964 206 resulting sequencing reads were assembled *de novo* in contiguous sequences using the default parameters in Canu (version 1.7) [38], with the exception of the expected genome size, which was set to the average genome size of members of the Tricholomataceae family deposited in GenBank (54.16 Mb). For diploid genomes with heterozygous regions such as the one from *D. pusillus* LMB4 (i.e. similar sections of a genome inherited from different parents), *de novo* assembly tools tend to create chimeric assemblies containing contigs from different haplotypes (i.e. sections of a genome inherited from the same parent). This results in highly fragmented assemblies that are artefactually too large in size. To simplify the search for genes encoding laccases and to avoid biasing general statistics such as genome size, assembly was reduced using Redundans (version 0.14a) [39]. This tool takes advantage of the long PacBio reads (where each read corresponds to the sequencing of a DNA strand) to find co-inherited genetic markers to generate single continuous homozygous regions. When ambiguity occurs in genome assembly, Redundans keeps the proper haplotype according to the quality of the heterozygous region.

## 2.11. Ligninolytic laccase annotations of the *D. pusillus* genome draft

Protein encoding genes were predicted with WebAUGUSTUS [40] using *Laccaria bicolor* as a training dataset. The resulting predicted gene sequences were annotated using the webserver of eggNOG-mapper [41]. Each putative laccase sequence was submitted to the Basic Local Alignment Search Tool for proteins BLASTp tool from the database at National Center for Biotechnology Information (NCBI) server to find a correlation with other laccase enzymes

reported on the Protein Data Bank server (PDB). Moreover, the four conserved copper-binding motifs, *i.e.* Cu1 (HWHGFFQ), Cu2 (HSHLSTQ), Cu3 (HPFHLHG), and Cu4 (HCHIDFHL) [42], were searched into these putative protein sequences. Also, sequences corresponding to putative laccases were further analyzed using InterProScan [43] to verify the presence of multicopper oxidase signatures (PS00079 and PS00080 Prosite entries, ExPASy Bioinformatics Resource Portal) and Cu-oxidase Pfam domains (PF00394, PF07731, and PF07732 entries) [44]. Comparisons with the Laccase and Multicopper Oxidase Engineering Database [45] was also used to validate that the identified sequences were laccases. Finally, this Whole Genome Shotgun project was deposited at DDBJ/ENA/GenBank under the accession QVIE00000000.

## 3. Results and discussion

### 3.1. Fungi isolation

From all fruit bodies collected, twelve axenic cultures were obtained and thirty one isolates exhibited fungal contamination from biota mycoparasitism associated to basidiomycetes, mainly from *Trichoderma* species. These fungi possess fungicide and antagonistic activity against basidiomycete cell walls, in addition to releasing enzymes such as chitinases and glucanases [46,47]. Moreover, basidiomycete recovery from collected samples can also suffer from competition with ascomycete fungi. Competition between these two fungi heavily relies on nutrient accessibility, growth factors favoring ascomycetes due to their faster growing pace in complete culture media, or even the presence of simple nutrient sources observed in advanced stages of wood decay [48]. Based on fruiting body macroscopic properties (front and back surface, color, texture, border margins, heights and widths), twelve fungi isolates were identified. Isolated strains belong to the orders i) *Hymenochaetale*: *Hyphodontia* (2 isolates), *Phellinus* (1 isolate); ii) *Polyporales*: *Aleurodiscus* (1 isolate), *Mycoacia* (2 isolates), *Stereum* (1 isolate), *Trametes* (1 isolate), *Tyromyces* (1 isolate), and *iii*) *Agaricales*: *Dictyopanus* (1 isolate), *Pleurotus* (2 isolates). Such orders are associated with oxidoreductase and hydrolase producers that cluster in the same evolutionary taxa (class *Agaricomycetes*). It is also worth mentioning that those fungi represent the most cited ligninolytic enzyme producers [49–51].

### 3.2. Screening of isolates

Fungal enzymatic extracts were screened for enzymes known to participate in the delignification process, i.e. laccases, manganese peroxidases, and lignin peroxidases. From the crude fungal enzymatic extracts obtained by SSF, only five isolates exhibited laccase activity in our screening assay. Surprisingly, we were unable to measure peroxidase activity other than through the ABTS assay. Since peroxidases are common enzymes present during fungi-catalyzed wood decay, peroxidase activity was either negligible in our isolates or the enzymatic assay was not sensitive enough to quantify such activity. It has been reported that variations in the concentrations of lignin, carbon, nitrogen, and the presence of chemical compounds such as inducers in the culture media could affect the profile of ligninolytic enzymes expressed and secreted during fermentation [52–54]. While current experiments cannot explain whether the lack of peroxidase activity is related to the composition of the culture media, the abovementioned results confirm previous reports suggesting that laccase activity is the most prevalent ligninolytic activity observed during fermentation with lignocellulose as substrate [55,56].

Isolates exhibiting ligninolytic activity were identified as *Dictyopanus* sp. LMB4 (22.3 U.L$^{-1}$), *Pleurotus* sp. LMB2 (69.5 U.L$^{-1}$), and *Pleurotus* sp. LMB3 (57.2 U.L$^{-1}$) (Fig 1). Laccase activity of the *Hyphodontia* and *Trametes* isolates was considered too low to warrant further characterization. For the three most active isolates, the highest laccase activity was detected

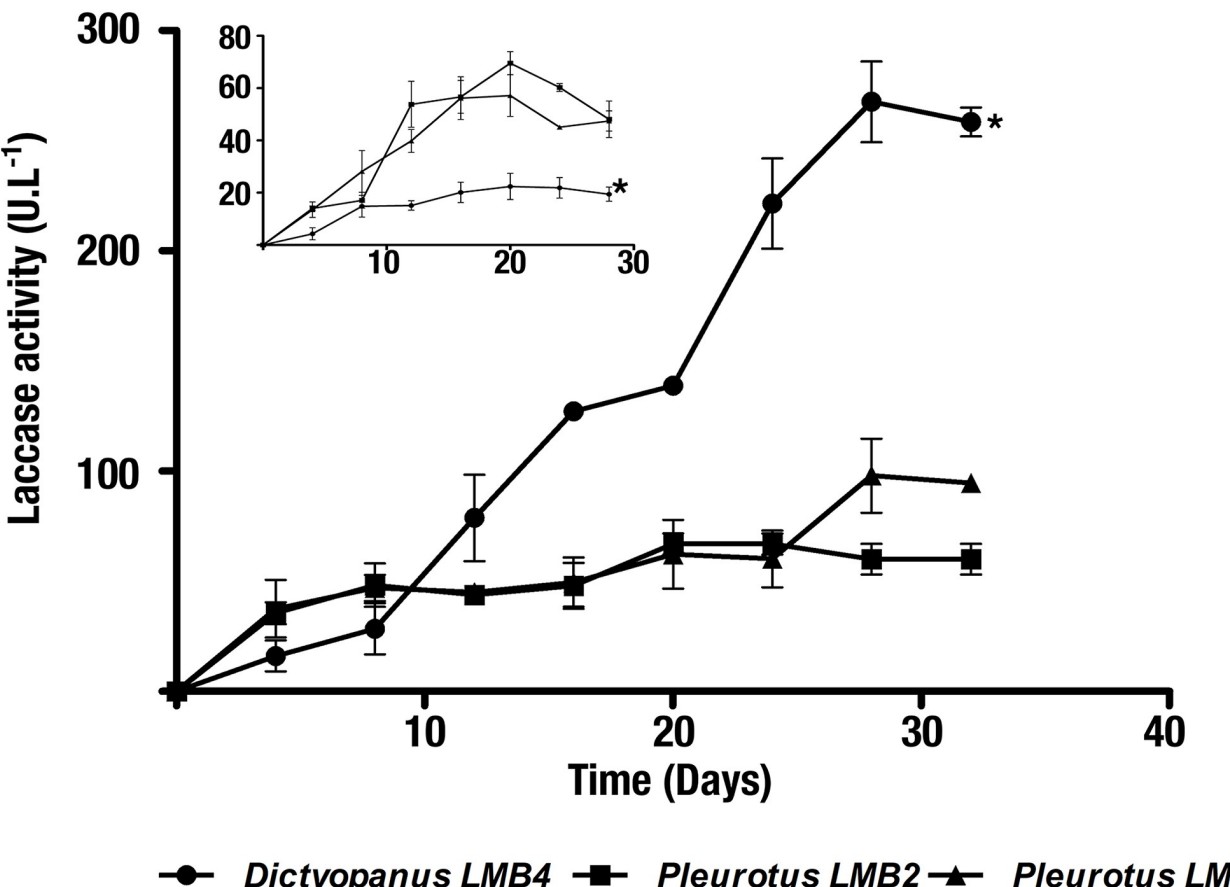

**Fig 1. Laccase activity of SSF isolates.** ABTS oxidation activity was tested for three culture supernatants from *Dictyopanus* LMB4 (circles), *Pleurotus* LMB2 (squares), and *Pleurotus* LMB3 (triangles) isolates. With a C/N ratio of 1.9 and in the absence copper, the *Pleurotus* spp. isolate exhibited the highest laccase activity (see inset). However, a 12-fold increase in laccase activity was observed in the *Dyctiopanus* sp. isolate with a 10-fold increase in the carbon-to-nitrogen ratio (19 C/N) and 5 mM copper (main histogram). Axes and units are the same for both histograms. The *Dictyopanus* LMB4 isolate is highlighted by an asterisk in both histograms.

after 20 days of fermentation. Using these 3 isolates, laccase activity conditions were optimized by increasing copper concentration and carbon-to-nitrogen ratios (C/N) [57,58]. As a result, the isolate exhibiting the highest laccase activity under these newly optimized conditions was *Dictyopanus* sp. LMB4 (267.6 U.L$^{-1}$ after 28 days of fermentation). To the best of our knowledge, this represents the first observation of significant laccase activity in a crude enzymatic extract from a *Dictyopanus* fungus. Furthermore, this activity is similar to a previously reported *Trametes* sp. laccase activity evaluated under comparable fermentation conditions using lignocellulosic by-products from oil palm (218.6 U.L$^{-1}$) [59]. The maximal laccase activities of the *Pleurotus* isolates were at least 5 times lower than the one observed in *Dictyopanus* sp. LMB4, with 98 U.L$^{-1}$ for *Pleurotus* sp. LMB2, and 66.9 U.L$^{-1}$ for *Pleurotus* sp. LMB3 (Fig 1).

Upon growth condition optimization, the crude enzymatic activity of *Dictyopanus* sp. LMB4 increased 6- and 12-fold after 20- and 28-day incubation, respectively, highlighting the importance of copper and carbon source accessibility for proper enzyme expression. The increase in laccase activity for fungal enzymatic extracts upon copper and glucose addition has been reported for *Colorios versicolor* and *Ganoderma lucidum*. These reports suggested that copper and glucose could respectively stimulate laccase expression and mycelial growth, further correlating with a proportional increase in the amount of laccase secreted by the fungi

[60,61]. For the enzymatic extract of *D. pusillus*, the calculated laccase activity obtained per gram of oil palm by-products was 31.5 U.g$^{-1}$ after 12 days of SSF. It is worth mentioning that this activity is four times higher than the previously reported laccase activity of a *Pycnoporus sanguineus* enzymatic extract obtained under similar SSF conditions using empty fruit bunch as substrate (7.5 U.g$^{-1}$) [62].

### 3.3. Molecular identification of *Dictyopanus* sp

In contrast to most organisms genetically identified using 16S ribosomal RNA sequencing, Internal Transcribed Spacer regions (ITS) is considered a more appropriate method to identify species in the fungi kingdom [63]. In the past, mycologists have used an arbitrary sequence similarity cutoff ranging between 3–5% ITS identity as a threshold for species differentiation. However, the natural variability of ITS sequences at the phylum level within the fungi kingdom complicates the use of such cutoff [63]. For instance, in Basidiomycota (to which the *Dictyopanus* genus belongs), the infraspecific ITS variability was reported to average at 3.3%, with a standard deviation of 5.62% [63]. This significantly limits the use of GenBank BLAST searches as the only source to properly identify fungi species, especially considering the fact that more than 27% of ITS sequences were submitted with insufficient taxonomic identification [64]. In addition, until 2003, nearly 20% of all fungal species listed in GenBank were incorrectly annotated [65]. As a result, using BLAST searches to categorize fungal species can lead to serious misidentification and characterization. Consequently, fungal specimen identification requires a careful, systematic, and multi-source process.

To overcome some of these limitations, we first performed preliminary *in situ* morphological identification of the samples collected in the Colombian forest. Genus level inspection was performed in the laboratory using macroscopic and microscopic examination, followed by final phylogenetic identification through DNA extraction and sequencing of ITS regions 1 and 4 [28]. This allowed identification of the *pusillus* species, to which the *Dictyopanus* LMB4 fungus sample belongs (Fig 2). The same analysis also allowed us to differentiate the evolutionary history for some members of the *Panellus* genus, with which members of the *Dictyopanus* genus are often confused. Results presented in Fig 2 support the usefulness of taxonomic classification performed during fungi sample collection, selection, and isolation. The *Dictyopanus* genus belongs to the *Agaricomycetes* class, and its genus is known to include species capable of bioluminescence, which have been suggested to be linked to delignification processes through the use of secondary compounds produced during lignin degradation [66]. *Dictyopanus* isolates were also reported as an alternative for the pretreatment of remazol brilliant blue R [67] and bamboo in ethanol production [68], further supporting the potential use of this fungus in large-scale biomass degradation.

### 3.4. Effect of pH and temperature on the fungal enzymatic extracts obtained from *D. pusillus*

Characterization of crude fungal enzymatic extracts showed that pH values between 3 and 5 provided the highest laccase activity for *D. pusillus* LMB4, with a maximum activity of 2,277 ± 36 UI*L$^{-1}$ at pH 3 (Fig 3A). This pH range corresponds to other laccase preferences in fungi [69]. Moreover, thermal stability of the crude *D. pusillus* LMB4 enzymatic extract was found to be quite robust, with reduced activity only observed at 60˚C (46 ± 5% activity loss after 6 hours of incubation). This behavior is quite different from that observed with the *T. versicolor* commercial laccase under the same experimental conditions, showing 28 ± 4% and 88 ± 2% activity loss after a 6h incubation at 50˚C and 60˚C, respectively (Fig 4). Thus, *D. pusillus* LMB4 appears to express laccases with enhanced thermostability and high tolerance

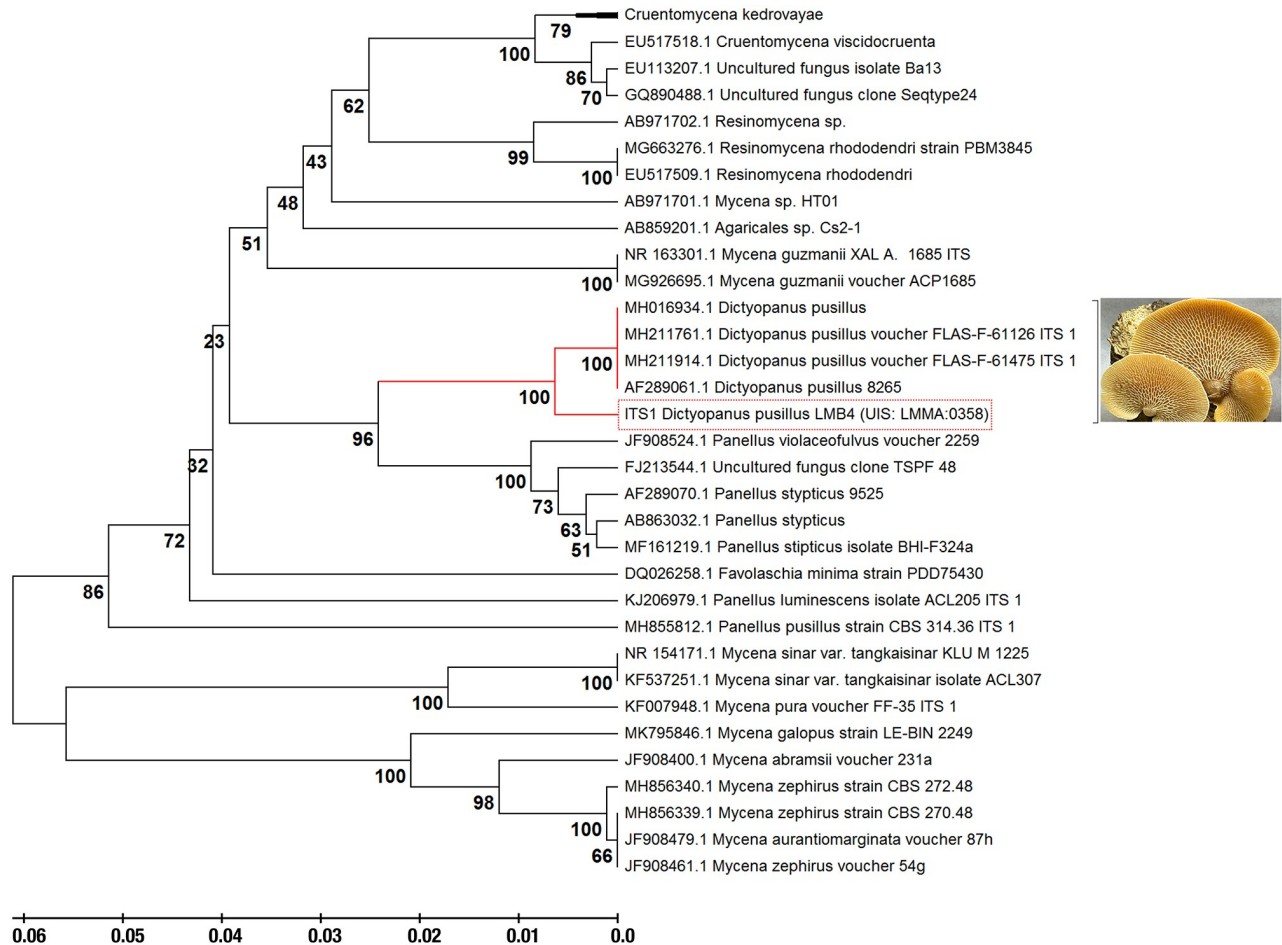

**Fig 2. Phylogenetic analysis of the pre-identified isolates labeled as *Dictyopanus* sp.** We used the ITS region 1 as the genetic marker to infer the evolutionary history of this fungus using the UPGMA protocol (see Materials and methods for details). The optimal tree analysis shows a branch length of 0.60, with clustering of species after a bootstrap of 500 replicates using the Maximum Composite Likelihood method to obtain evolutionary distances between members. The species was identified as *Dictyopanus pusillus*. The phylogenetic tree was drawn to use the same branch length units as those of the evolutionary distances. This analysis was performed using the standalone MEGA software, version 10.0.5.

to low pH values. However, long incubation of this crude fungal enzymatic extract at low pH resulted in an important activity loss of 80.1 ± 0.2% after two hours of incubation (Fig 3B). Previous studies have shown that a laccase from *Physisporinus rivulosus* remained stable at 50°C with optimal activity at pH 3.5 [70]. Similarly, a laccase from *Trametes trogii* was shown to sustain temperatures up to 75°C, although only for short 5-min incubations [71]. Nevertheless, our results suggest that the laccase activity from the *D. pusillus* LMB4 extract has higher tolerance to acidic and thermally induced perturbations than previously identified fungal laccases.

## 3.5. Using *D. pusillus* for the simultaneous pretreatment and saccharification of empty fruit bunch

Fiber analysis of palm empty fruit bunch revealed a composition of 77.53% NDF, 58.32% ADF, and 17.15% ADL (see Materials and methods for details). These values indicate that empty fruit bunch composition of the lignocellulosic polymer used for SSF was 40.79% cellulose, 19.21% hemicellulose, 17.15% lignin, and 22.47% impregnated oil and ashes. These results are in accordance with typically reported empty fruit bunch composition, with cellulose being

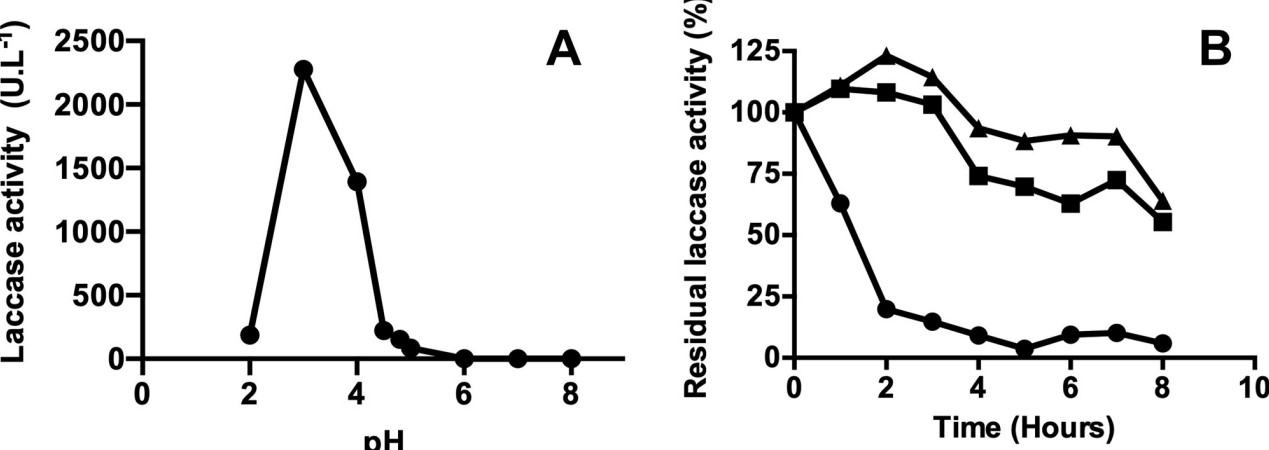

**Fig 3. pH tolerance of a *D. pusillus* LMB4 extract exhibiting laccase activity.** A) Laccase activity from a crude *D. pusillus* LMB4 enzymatic extract at different pH values. B) pH stability assay performed with the crude enzymatic extract from *D. pusillus* LMB4. Laccase activity was evaluated at 40°C under different pH conditions: pH 3 (circles), pH 4 (squares), and pH 5 (triangles). Average values and standard deviation were obtained from three replicates of each point.

the main component, followed by hemicellulose and lignin [72]. Reducing sugar release was observed when the cellulolytic enzymatic extract from *T. reesei* was used alone ($20.84 \pm 0.7$ g.g$^{-1}$). Higher reducing sugar release from empty fruit bunch was also observed when the cellulolytic enzymatic extract from *T. reesei* was used with the commercial laccase enzyme from *T. versicolor* ($46.47 \pm 5.9$ g.g$^{-1}$) or the enzymatic extract from *D. pusillus* ($44.80 \pm 5.2$ g.g$^{-1}$), confirming that ligninolytic enzymes such as laccases favor cellulose hydrolysis, as previously reported [73,74]. These results suggest that a combination of cellulolytic and ligninolytic enzymes enhance the release of reducing sugars. However, production of reducing sugars was not significantly different when the commercial laccase from *T. versicolor* or enzymatic extracts from *D. pusillus* were mixed with the cellulolytic enzymatic extract from *T. reesei* (Fig 5).

To identify the dominant variables affecting reducing sugar release, we compared the effects of pH, temperature, copper concentration, and laccase (U.L$^{-1}$) or cellulase (UPF) concentration using a multilevel factorial experimental design ($P = 0.05$ with a confidence level of 95%) (Fig 6). For the pretreatment and saccharification experiment with laccase from *T. versicolor*, pH ($P<0.0001$), temperature ($P = 0.0006$), and cellulases ($P = 0.0021$) were the three dominant variables affecting activity. For pretreatment and saccharification with the enzymatic extract from *D. pusillus*, pH ($P<0.0001$) was the only dominant variable affecting enzymatic performance (Fig 6). Our results indicate that pH values between 3–4 and temperatures up to 45°C promote sugar release by simultaneous pretreatment and saccharification. These results confirm what was observed in our stability experiments, where the activity of the enzymatic extract was almost obliterated at pH values higher than 4 (Fig 3A). It is worth mentioning that cellulase ($P = 0.0021$) is the third most important contributing factor to activity when simultaneous pretreatment and saccharification is performed with the commercial laccase (Fig 6A), a result we do not observe with the enzymatic extract from *D. pusillus* (Fig 6B). The requirement of a cellulase activity in the case of the commercial laccase are perhaps due to the combined production of ligninolytic and cellulolytic enzymes in the basidiomycete fungi during wood decay processes [75,76]. Some authors have also demonstrated the efficiency of enzymatic extracts from basidiomycetes for the SSF production of ligninolytic and cellulolytic enzymes

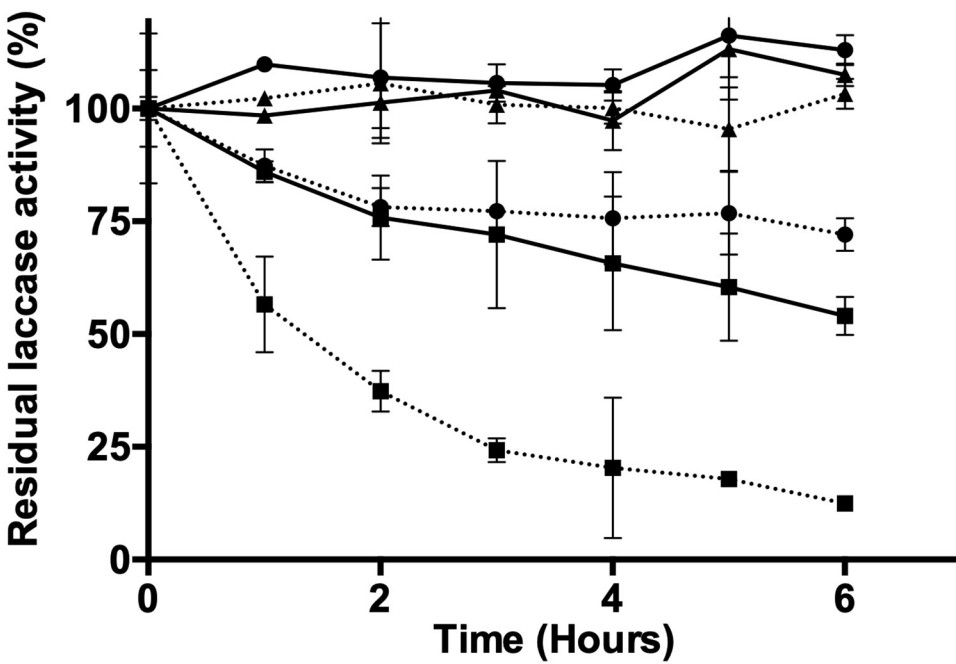

**Fig 4. Thermal stability of a *D. pusillus* LMB4 extract exhibiting laccase activity.** Laccase activity was measured after different temperature incubations: 40°C (triangles), 50°C (circles), and 60°C (squares). Solid lines represent the crude fungal enzymatic extract from *D. pusillus* LMB4, while dashed lines represent the commercial laccase from *T. versicolor*, 53739. Average values and standard deviation were obtained from three replicates of each point.

using wheat straw as substrate [77]. These results further highlight the importance of *D. pusillus* as an efficient, accessible, and economical source of relevant biotechnological assets in the field of delignification processes.

The highest reducing sugar concentration obtained with the enzymatic extract of *D. pusillus* was 65.87 g.g$^{-1}$ (pH 4.5, 45°C, 2:1 laccase-to-cellulase ratio). In the same conditions, reducing sugar production reached 64.13 g.g$^{-1}$ for the commercial laccase from *T. versicolor*. These results confirm that the enzymatic extract from *D. pusillus* exhibits similar ligninolytic efficiency than the purified commercial laccase from *T. versicolor*. Additionally, empty fruit bunch represent a good lignocellulose source for reducing sugar production since palm oil bunches are subjected to a first round of "sterilization" to extract oil palm fruits from the bunch, which effectively acts as a pretreatment during palm oil extraction. As a result, this initial pretreatment might improve the delignification process performed by the enzymes. Lignocellulose breakdown of empty fruit bunch and empty fruit bunch pulp was previously reported using the white rot fungi *T. versicolor* TISTR 3224, *Phanerochaete chrysosporium* CECT 2798, and *Pleurotus ostreatus* CEC20311. These fungi were also used as efficient pretreatments for lignin removal in empty fruit bunch [78,79]. To the best of our knowledge, only one study reported the use of fungal enzymatic extracts with laccase activity from *Pycnoporus sanguineus* UPM4 as a pretreatment of empty fruit bunch to increase production of reducing sugars in similar conditions [80]. This report and the results presented here on the use of a crude fungal enzymatic extracts exhibiting laccase activity from a white-rot fungi reinforce the relevance of using ligninolytic enzymatic extracts as a valuable tool for the pretreatment of lignocellulose in empty fruit bunch.

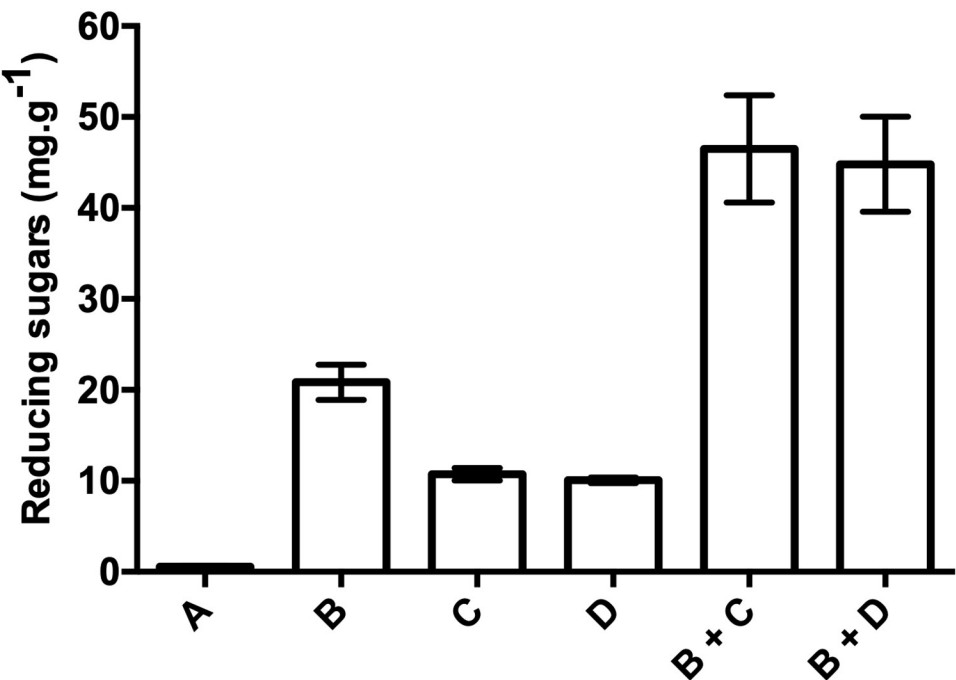

**Fig 5. Comparative production of reducing sugars from empty fruit bunch using fungal enzymatic extracts alone or in combination.** Reducing sugar release was observed: A) without any fungal enzymatic extract, B) with a cellulolytic extract from *T. reesei*, C) with a commercial laccase from *T. versicolor*, D) with the enzymatic extract from *D. pusillus*. Combinations of B+C and B+D were also tested. Standard deviation was obtained from three replicates in each condition.

## 3.6. Genome sequencing of *D. pusillus* LMB4 and laccase sequence annotation

Given the striking ligninolytic activity of *D. pusillus* LMB4 and lack of genomic data available to identify and compare potential enzyme homologs promoting such activity in this organism, we used long-reads single-molecule real-time technology (PacBio) to perform genomic DNA sequencing of *D. pusillus* LMB4. This allowed analysis and annotation of a number of putatively encoded laccases in this genome, offering means to predict potential enzymes involved in this ligninolytic activity. After *de novo* assembly of the genome from *D. pusillus* LMB4, we estimated heterozygosity at 13.53% using the Redundans tool (i.e. similar sections of a diploid genome, but inherited from different parents). Comparing this value to genome heterozygosity in nearby fungi remains difficult due to the lack of reported information, namely for the order Agaricales. However, a previous study reporting on the sequence of 90 fungi suggested that genomes of members of the phylum Basidiomycota, of which *D. pusillus* belongs, typically have high levels of heterozygosity [81]. Reduction in homozygous genome allowed the assembly of 49.37 Mbp distributed in 3463 contigs (N50 = 23,741 bp) (Table 1). After splicing of the 95,174 annotated introns, a total of 16,866 coding sequences (CDSs) were predicted to be encoded in the genome of *D. pusillus* LMB4. Of this number, we confidently annotated a total of 13 CDSs as complete putative laccase sequences, which were further aligned with a previously reported laccase homolog from *Trametes* to identify consensus regions and conserved motifs (Fig 7, S1 Table in S1 File). Our results show that all putative laccases encoded in the *D. pusillus* genome preserve the four conserved copper-binding motifs normally observed in this enzyme family, *i.e.* Cu1 (HWHGFFQ), Cu2 (HSHLSTQ), Cu3 (HPFHLHG), and Cu4

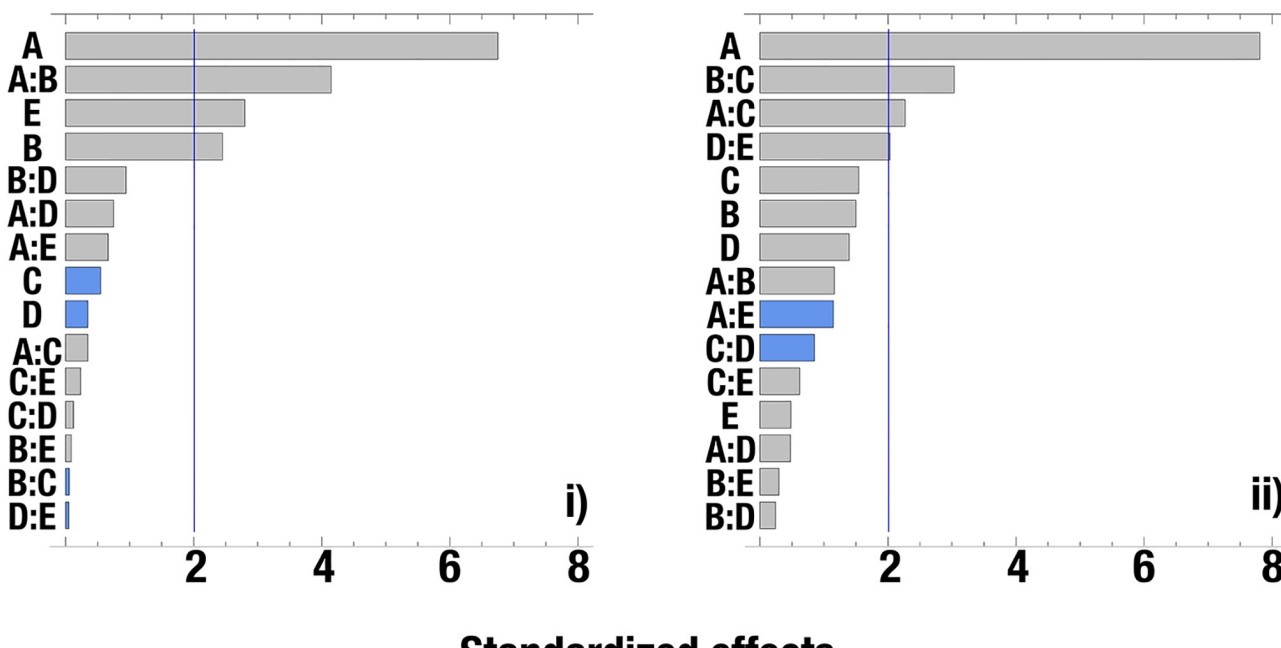

**Fig 6. Pareto charts from multilevel factorial experimental design analysis. i)** Cellulolytic extract from *T. reesei* mixed with commercial laccase from *T. versicolor*. **ii)** Cellulolytic extract from *T. reesei* mixed with enzymatic extract from *D. pusillus*. Parameters: A, pH; B, temperature; C, copper concentration; D, U.L$^{-1}$ of laccase, and E, UPF of cellulase. Vertical lines represent the statistically significant threshold of 95% confidence with a $P = 0.05$, while grey and blue bars highlight positive and negative effects, respectively.

(HCHIDFHL) [42]. These make them potentially promising candidates for future functional investigation of new laccases exhibiting interesting properties with respect to activity, stability, and industrial tolerance.

## 4. Conclusion

The present work demonstrates that a crude fungal enzymatic extract from a wild Colombian source of *D. pusillus* LMB4 exhibits significant laccase activity ($267 \pm 18$ U.L$^{-1}$). This crude fungal enzymatic extract was probed for the successful pretreatment of low-cost lignocellulosic raw materials (oil palm by-products), suggesting that an upscaling of this process could potentially help with the delignification of starting materials in cellulosic bioethanol production. An increase in copper and glucose concentration during solid-state fermentation proved beneficial, resulting in a 12-fold increase in laccase activity and suggesting that ligninolytic enzyme expression can further be induced to improve enzyme production in *D. pusillus* LMB4. The

**Table 1. Assembly of the *D. pusillus* LMB4 genome draft.**

| Feature | Value |
|---|---|
| Genome assembly size (Mbp) | 49.37 |
| Number of contigs | 3463 |
| N50 (bp) | 23,741 |
| GC (%) | 53.08 |
| Number of CDSs | 16,866 |
| Number of introns | 95,174 |
| Heterozygosity (%) | 13.53 |

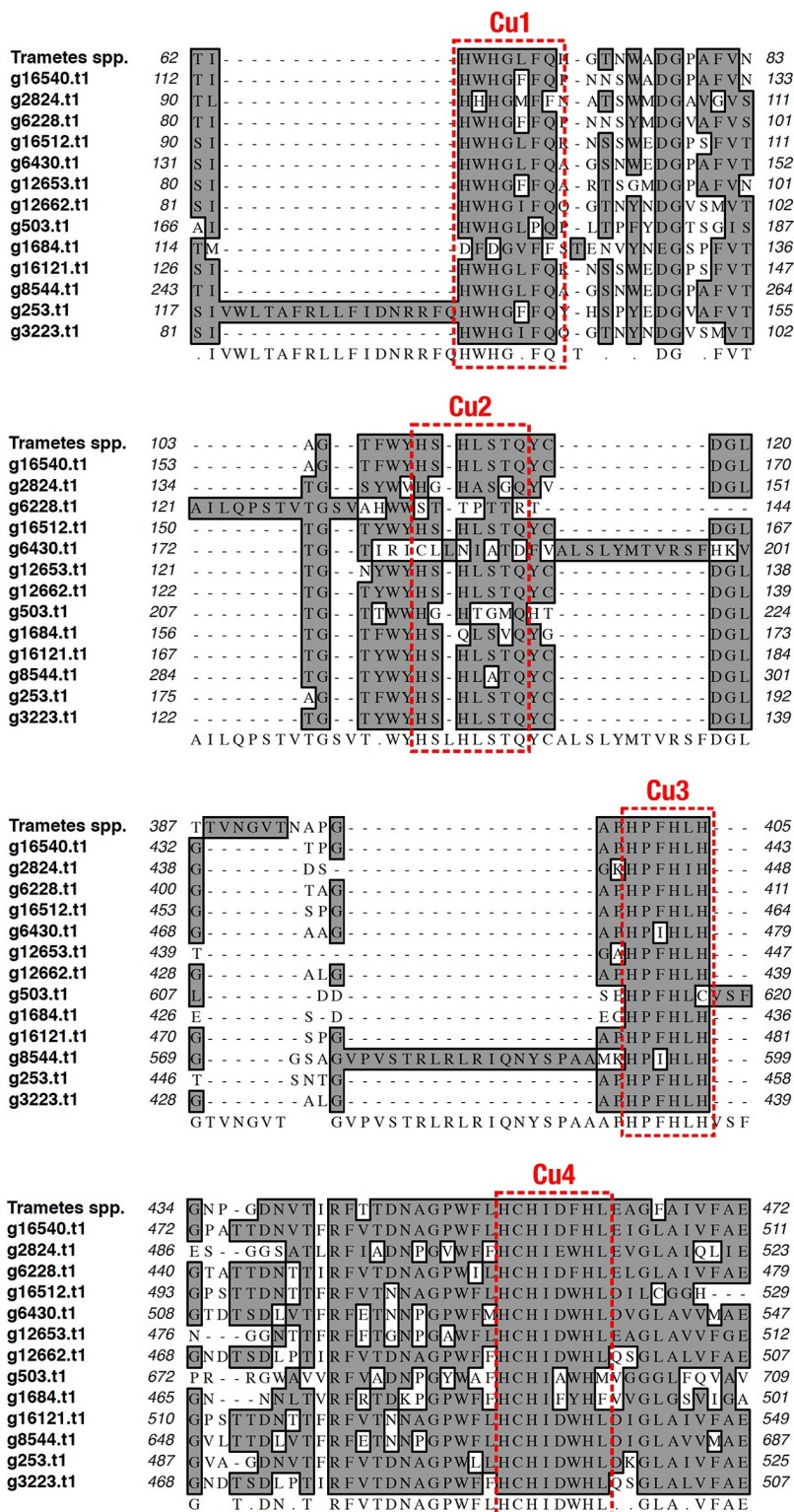

**Fig 7. Multiple Sequence Alignment (MSA) of the 13 putative laccases identified in the genome of *Dictyopanus pusillus* LMB4.** The four conserved copper-binding motifs are highlighted by red dashed rectangles. The laccase sequence of the *Trametes* genus was used as reference to perform the MSA. Consensus sequence is presented on the bottom of the alignment. Putative laccase genes are identified as in S1 Table in S1 File.

simultaneous pretreatment and saccharification of empty fruit bunch also illustrated that the enzymatic extract from *D. pusillus* exhibits good ligninolytic capacity at acidic pH, in addition to demonstrating higher pH and thermal stability than the purified commercial laccase from *T. versicolor*. These properties demonstrate the efficiency of such crude enzymatic extract from *D. pusillus* as a versatile biotechnological tool for lignocellulose pretreatment such as for cellulosic bioethanol production. Genome sequencing of *D. pusillus* LMB4 also revealed 13 laccases and a significant number of other putative enzymes that could be exploited and/or engineered to develop more efficient delignification pre-treatments. These results thus present the first few stages in the implementation of a strategy that combines genome data mining and computational modelling as efficient approaches to identify promising new protein engineering candidates as new sets of catalysts with application in delignification processes.

## Supporting information

**S1 File.**
(DOCX)

## Acknowledgments

This work was partially supported by a grant from Universidad Industrial de Santander, Vice-rrectoria de Investigación y Extension (Grant number 5199) (to C.I.S. and D.M.V.), Industrias Acuña INAL LTDA (Grant number 8712) (to C.I.S.), and a Natural Sciences and Engineering Research Council of Canada (NSERC), via Discovery Grant RGPIN-2016-05557 (to N.D.). A. T.V. received a Postdoctoral Fellowship from NSERC. A.M.R. was supported by a doctoral scholarship from the Colombian Departamento Administrativo de Ciencia, Tecnología e Innovación (Colciencias) (PhD scholarship 567, 2012), and was the recipient of a scholarship from the Emerging Leaders in the Americas Program (ELAP) from the Government of Canada. F.V. and N.D. hold Fonds de Recherche Québec-Santé (FRQS) Research Scholar Junior 1 and Senior Career Awards, respectively (numbers 35038 and 281993). There was no additional external funding received for this study.

## Author Contributions

**Conceptualization:** Andrés M. Rueda, Yossef López de los Santos, Antony T. Vincent, Sonia A. Ospina, Frédéric J. Veyrier, Nicolas Doucet.

**Data curation:** Andrés M. Rueda, Yossef López de los Santos, Antony T. Vincent, Myriam Létourneau, Clara I. Sánchez.

**Formal analysis:** Andrés M. Rueda, Yossef López de los Santos, Antony T. Vincent, Myriam Létourneau.

**Funding acquisition:** Clara I. Sánchez, Daniel Molina V., Frédéric J. Veyrier, Nicolas Doucet.

**Investigation:** Andrés M. Rueda, Yossef López de los Santos, Antony T. Vincent, Myriam Létourneau, Nicolas Doucet.

**Methodology:** Andrés M. Rueda, Yossef López de los Santos, Antony T. Vincent, Myriam Létourneau, Inés Hernández, Clara I. Sánchez.

**Project administration:** Inés Hernández, Sonia A. Ospina, Frédéric J. Veyrier, Nicolas Doucet.

**Resources:** Inés Hernández, Clara I. Sánchez, Frédéric J. Veyrier, Nicolas Doucet.

**Supervision:** Sonia A. Ospina, Frédéric J. Veyrier, Nicolas Doucet.

**Validation:** Andrés M. Rueda, Yossef López de los Santos, Antony T. Vincent, Myriam Létourneau, Frédéric J. Veyrier, Nicolas Doucet.

**Visualization:** Andrés M. Rueda, Yossef López de los Santos, Antony T. Vincent, Myriam Létourneau.

**Writing – original draft:** Andrés M. Rueda, Yossef López de los Santos, Antony T. Vincent, Nicolas Doucet.

**Writing – review & editing:** Andrés M. Rueda, Yossef López de los Santos, Antony T. Vincent, Myriam Létourneau, Frédéric J. Veyrier, Nicolas Doucet.

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
