## [Decision Letter · Decision Letter 0]

30 Mar 2020

PONE-D-19-35214

Genome sequencing and functional characterization of a Dictyopanus pusillus fungal extract offers a promising alternative for lignocellulose pretreatment of oil palm residues

PLOS ONE

Dear Dr. Doucet,

Thank you for submitting your manuscript to PLOS ONE. After careful consideration, we feel that it has merit but does not fully meet PLOS ONE’s publication criteria as it currently stands. Therefore, we invite you to submit a revised version of the manuscript that addresses the points raised during the review process.

The reviewers believe that you are addressing an important and timely topic.  However, all felt that the manuscript would benefit from significant revision. In particular, there were questions about statistical analyses and interpretation of some data. There were also concerns about organization of the manuscript, missing information and excessive use of abbreviations.  See the 3 reviews for more detailed suggestions.

We would appreciate receiving your revised manuscript by May 30, 2020. To enhance the reproducibility of your results, we recommend that if applicable you deposit your laboratory protocols in protocols.io, where a protocol can be assigned its own identifier (DOI) such that it can be cited independently in the future. For instructions see: http://journals.plos.org/plosone/s/submission-guidelines#loc-laboratory-protocols

We look forward to receiving your revised manuscript.

Kind regards,

Katherine A. Borkovich, Ph.D.

Academic Editor

PLOS ONE

Journal Requirements:

2.Thank you for stating in your Funding Statement:

"This work was partially supported by a grant from Universidad Industrial de Santander, Vicerrectoria de Investigación y Extension (Grant number 5199) (to C.I.S. and D.M.V.), Industrias Acuña LTDA, and a Natural Sciences and Engineering Research Council of Canada (NSERC), via Discovery Grant RGPIN-2016-05557 (to N.D.). A.T.V. received a Postdoctoral Fellowship from NSERC. A.M.R. was supported by a doctoral scholarship from the Colombian Departamento Administrativo de Ciencia, Tecnología e Innovación (Colciencias) (PhD scholarship 567, 2012), and was the recipient of a scholarship from the Emerging Leaders in the Americas Program (ELAP) from the Government of Canada. F.V. and N.D. hold Fonds de Recherche Québec-Santé (FRQS) Research Scholar Junior 1 and Junior 2 Career Awards, respectively (numbers 35038 and 32743)."

"This work was partially supported by a grant from Universidad Industrial de Santander, Vicerrectoria de Investigación y Extension (Grant number 5199) (to C.I.S. and D.M.V.), Industrias Acuña LTDA, and a Natural Sciences and Engineering Research Council of Canada (NSERC), via Discovery Grant RGPIN-2016-05557 (to N.D.). A.T.V. received a Postdoctoral Fellowship from NSERC. A.M.R. was supported by a doctoral scholarship from the Colombian Departamento Administrativo de Ciencia, Tecnología e Innovación (Colciencias) (PhD scholarship 567, 2012), and was the recipient of a scholarship from the Emerging Leaders in the Americas Program (ELAP) from the Government of Canada. F.V. and N.D. hold Fonds de Recherche Québec-Santé (FRQS) Research Scholar Junior 1 and Junior 2 Career Awards, respectively (numbers 35038 and 32743)."

We note that you received funding from a commercial source: Industrias Acuña LTDA

Please respond by return email with your amended Competing Interests Statement and we will change the online submission form on your behalf.

Reviewers' comments:

Reviewer's Responses to Questions

**Comments to the Author**

1. Is the manuscript technically sound, and do the data support the conclusions?

Reviewer #1: Yes

Reviewer #2: Partly

Reviewer #3: Yes

2. Has the statistical analysis been performed appropriately and rigorously? 

Reviewer #1: N/A

Reviewer #2: I Don't Know

Reviewer #3: Yes

3. Have the authors made all data underlying the findings in their manuscript fully available?

Reviewer #1: Yes

Reviewer #2: Yes

Reviewer #3: Yes

4. Is the manuscript presented in an intelligible fashion and written in standard English?

Reviewer #1: Yes

Reviewer #2: No

Reviewer #3: Yes

5. Review Comments to the Author

Reviewer #1: The paper entitled “Genome sequencing and functional characterization of a Dictyopanus pusillus fungal extract offers a promising alternative for lignocellulose pretreatment of oil palm residues” gives an insight into the isolation, identification of potent ligninolytic enzyme producing strains form different isolates, functional characterization and genomic study of the highly potent strains was performed. The manuscript is nicely written and is technically sound.

However, there are certain points which the authors need to explain or revise before the paper may be accepted for publication. Author must carefully revise the MS in light of below given comments.

General Comments

1. Throughout the text phrase“fungal enzymatic extract” is used in title it’s just fungal extract please check if the “fungal enzymatic extract” can be used.

2. In Page no 4 line “We found that laccase activity was the main enzymatic contributor to our fungal extracts, which included a highly active isolate from D. pusillus LMB4.” must be moved to the result section

3. Please elaborate the methodology for preparation of wheat bran extract

4. In Page 17-18, “These results are also in agreement with prior observations suggesting that basic pH is a desirable property for laccase used in biotechnological processes, since low pH values were linked to increased enzyme degradation” The authors are suggesting basic pH has positive effect where the results shows otherwise. The enzyme is highly active in pH range of 3-4 during production (Fig 3A) and produced enzyme was highly stable in pH range of the 4 and 5 (Fig 3B) . If possible the author must try higher pH then 5 such as 7-9 for claiming enzyme to be active in basic environment or must stick to the observations obtained and refrain from using such statements contradicting their results.

5. In fig 6 components A, B, C, D, E were explained in figure title but what does each image suggest is not represented . You have used small “a” and “b” in figure to denote both figure however in the marking for table it is denoted by capital A and B further capital A, B was used to represent different parameters Its is confusing for reader so please represent what does fig a) and b) or may use Fig 6 (i) or (ii)

6. Regarding the submitted manuscript to adhere with author guidelines of the MS. (Reviewer have checked with recently published paper in PloSOne)

(a) Results is published separately from discussion but in present paper the Resutls and Discussion is one section, Please check with author guidelines once

(b) Referencing pattern in the text is not is in line with the PlosOne

Please check and revise manuscript after meeting the author guideline

Reviewer #2: The information on genome sequencing is scanty. The statistical data/ p values in Figures 3, 4, 5 are missing. Information on statistical optimization cellulolytic digestion is not available. The manuscript can be perhaps split into two seperate manuscripts with more relevant data and sent for publication.

Reviewer #3: The manuscript submitted by Rueda et al. focuses on biomass delignification using fungal strain to do pretreatment, the topic is very important, since biomass utilization, especially the agriculture waste ,including the oil palm residues the authors are interested in, is still a problem at the moment.

The study isolates a strain of Dictyopanus pusillus with highly laccases activity, and the author also show the strain they identified could be a good candidate for biomass (the oil palm residues here) pretreatment. The released sugar got significantly increased with the fungus pretreatment compare the without fungus.

In order to make the paper more easy reading and understanding, I have some suggestions before the paper could be accepted for publication.

1.It would be better to have a figure to show the whole experiment design, to show the whole process of the study, including fungal strain isolation, sequencing, enzymes determination, biomass pretreatment and so on.

2.There are a lot of uncommon abbreviations, such as EFB, SPS, very hard to following, it is better just write the full name.

3.In the page 19, the author said, the genome heterozygosity at 13,53%, please explain more about this heterozygosity, not every reader are familiar what this really means, it come from the sampling or it should be, because this fungus strain…

6. PLOS authors have the option to publish the peer review history of their article (what does this mean?). If published, this will include your full peer review and any attached files.

Reviewer #1: No

Reviewer #2: No

Reviewer #3: No

---

## [Author Response · Author response to Decision Letter 0]

1 Jun 2020

Montreal, May 30, 2020

Katherine A. Borkovich, Ph.D. 

Academic Editor

PLOS ONE 

Re: Revision of manuscript ID FJ-18-0974

Dear Prof. Borkovich,

Thank you for sending the reviews of the manuscript entitled “Genome sequencing and functional characterization of a Dictyopanus pusillus fungal enzymatic extract offers a promising alternative for lignocellulose pretreatment of oil palm residues” by Andrés M. Rueda, Yossef López de los Santos, Antony T. Vincent, Myriam Létourneau, Inés Hernández, Clara I. Sánchez, Daniel Molina V., Sonia A. Ospina, Frédéric J. Veyrier and Nicolas Doucet. 

We have revised the manuscript based on suggestions and recommendations provided by the reviewers. We thank you and the reviewers for taking the time to provide helpful suggestions to make our manuscript stronger. We hope you and the reviewers will find our revised manuscript suitable for publication. 

Best regards,

Nicolas Doucet, Ph.D. 

Professor

 

Comments from the Editor

1. Please provide an amended statement that declares *all* the funding or sources of support (whether external or internal to your organization) received during this study. (…) Please also include the statement “There was no additional external funding received for this study.” in your updated Funding Statement. Please include your amended Funding Statement within your cover letter. We will change the online submission form on your behalf. 

Response: Here is our updated Funding Statement:

This work was partially supported by a grant from Universidad Industrial de Santander, Vicerrectoria de Investigación y Extension (Grant number 5199) (to C.I.S. and D.M.V.), Industrias Acuña INAL LTDA (Grant number 8712) (to C.I.S.), and a Natural Sciences and Engineering Research Council of Canada (NSERC), via Discovery Grant RGPIN-2016-05557 (to N.D.). A.T.V. received a Postdoctoral Fellowship from NSERC. A.M.R. was supported by a doctoral scholarship from the Colombian Departamento Administrativo de Ciencia, Tecnología e Innovación (Colciencias) (PhD scholarship 567, 2012), and was the recipient of a scholarship from the Emerging Leaders in the Americas Program (ELAP) from the Government of Canada. F.V. and N.D. hold Fonds de Recherche Québec-Santé (FRQS) Research Scholar Junior 1 and Senior Career Awards, respectively (numbers 35038 and 281993). There was no additional external funding received for this study.

2. Please provide an amended Competing Interests Statement that explicitly states this commercial funder. (…) Within this Competing Interests Statement, please confirm that this does not alter your adherence to all PLOS ONE policies on sharing data and materials by including the following statement: "This does not alter our adherence to PLOS ONE policies on sharing data and materials.” Please respond by return email with your amended Competing Interests Statement and we will change the online submission form on your behalf.

Response: Competing Interests: Private funding provided by Industrias Acuña INAL LTDA (Grant number 8712) resulted from an internal competition by the Universidad Industrial de Santander (UIS) to support part of the current research project performed in the laboratory of Clara I. Sánchez at UIS (Colombia). Industrias Acuña INAL LTDA does not commercially benefit from the publication of this manuscript, nor was it involved in the conception, design, processing, analysis, or communication of the results and research project. All authors are fully committed to the full disclosure of all results and research materials resulting from the current research. All authors proclaim that this declaration of competing interests does not alter their adherence to all the PLOS ONE policies on sharing data and material.

Response: These data were originally presented in a complementary fashion and were not essential for the understanding of the core research presented in this work. Consequently, we have removed the corresponding phrase. 

4. Please include captions for your Supporting Information files at the end of your manuscript, and update any in-text citations to match accordingly.

Response: As requested, we have added captions for our Supporting Information files at the end of our manuscript. In-text citations were verified and correctly cited.

Comments from Reviewer 1.

1. Throughout the text phrase “fungal enzymatic extract” is used in title it’s just fungal extract please check if the “fungal enzymatic extract” can be used.

Response: We thank the reviewer for highlighting this discrepancy. For clarity and consistency, we have updated the text to include “fungal enzymatic extract” in all previously ambiguous instances that did not immediately relate to D. pusillus or any other fungi listed in each context, including in the title of the manuscript. 

2. In Page no 4 line “We found that laccase activity was the main enzymatic contributor to our fungal extracts, which included a highly active isolate from D. pusillus LMB4.” must be moved to the result section

Response: Respectfully, we believe it is important for a reader to get a general idea of the main conclusions and results of the manuscript at the end of the introduction. We consider that this sentence is sufficiently broad and general to encompass this overall idea. We obviously discuss these data in more detail in the Results sections.

3. Please elaborate the methodology for preparation of wheat bran extract

Response: In response to this request, we have clarified and extended our methodological description of wheat brand extraction and fungi isolation in Section 2.1: “Isolation of the collected fungi was performed in wheat bran extract agar composed of 18 g.L-1 agar, 10 g.L-1 glucose, 5 g.L-1 peptone, 2 g.L-1 yeast extract, 0.1 g.L-1 KH2PO4, 0.1 g.L-1 MgSO4.7H2O, 0.085 g.L-1 MnSO4, 0.1 g.L-1 chloramphenicol, 0.1 g.L-1, 600 U.L-1 nystatin, and 1000 mL wheat bran extract. Wheat bran extract was obtained by filtering 175 g.L-1 of wheat brand soaked in distilled water for 1 h.” 

4. In Page 17-18, “These results are also in agreement with prior observations suggesting that basic pH is a desirable property for laccase used in biotechnological processes, since low pH values were linked to increased enzyme degradation.” The authors are suggesting basic pH has positive effect where the results shows otherwise. The enzyme is highly active in pH range of 3-4 during production (Fig 3A) and produced enzyme was highly stable in pH range of the 4 and 5 (Fig 3B). If possible the author must try higher pH then 5 such as 7-9 for claiming enzyme to be active in basic environment or must stick to the observations obtained and refrain from using active in basic environment or must stick to the observations obtained and refrain from using such statements contradicting their results.

Response: This was a significant proofreading oversight on our part and we thank the reviewer for pointing out this contradiction in our analysis. We have removed these sentences, re-analyzed and re-written Section 3.4 relating to Figure 3 to more appropriately describe our observations, in addition to including quantifiable data and statistical analyses. We have also updated the quality and readability of Figures 3, 4 and 5.

5. In Fig 6 components A, B, C, D, E were explained in figure title, but what does each image suggest is not represented. You have used small “a” and “b” in figure to denote both figure; however in the marking for table, it is denoted by capital A and B. Further capital A, B was used to represent different parameters. It is confusing for a reader, so please represent what does fig a) and b) or may use Fig 6 (i) or (ii)

Response: We agree with the reviewer that panel numbering and identification was confusing in this figure. Consequently, we made a new version of this figure with renamed panels i) and ii). We also standardized and improved font usage and consistency in accordance with other figures.

Comments from Reviewer 2.

1. The information on genome sequencing is scanty.

Response: We apologize for the lack of precision in our original description of the genome sequencing and assembly methods. We have extended Sections 2.10 and 3.6 to clarify this, in addition to providing additional methodological and analyzing details.

2. The statistical data/p values in Figures 3, 4, 5 are missing. (…) Information on statistical optimization cellulolytic digestion is not available.

Response: We have clarified statistical data calculations in the legends of Figures 3, 4 and 5. We have also better outlined and described that the fungal enzymatic extracts exhibiting laccase activity and cellulases were subjected to a multilevel factorial experimental design (3 levels with 5 variables) to evaluate significant variables in the experimental process. This description is now better highlighted, cited and discussed in Sections 2.9 and 3.5. As requested, we also present all statistical p-values.

Comments from Reviewer 3.

1. It would be better to have a figure to show the whole experiment design, to show the whole process of the study, including fungal strain isolation, sequencing, enzymes determination, biomass pretreatment and so on.

Response: We thank the reviewer for this great suggestion. We have built a new experimental design figure that provides overall information on the strategy and procedures that were used in the current study. This is now Figure S1. 

2. There are a lot of uncommon abbreviations, such as EFB, SPS, very hard to following, it is better just write the full name.

Response: As requested, we have removed all EFB and SPS abbreviations and replaced them with their respective full descriptions.

3. In the page 19, the author said, the genome heterozygosity at 13,53%, please explain more about this heterozygosity, not every reader are familiar what this really means, it come from the sampling or it should be, because this fungus strain...

Response: In addition to expanding our explanation of genome sequencing methodology and analysis, we have also clarified the genome heterozygosity meaning in Section 2.10 and discussion in Section 3.6.

---

## [Decision Letter · Decision Letter 1]

19 Jun 2020

Genome sequencing and functional characterization of a Dictyopanus pusillus fungal enzymatic extract offers a promising alternative for lignocellulose pretreatment of oil palm residues

PONE-D-19-35214R1

Dear Dr. Doucet,

We’re pleased to inform you that your manuscript has been judged scientifically suitable for publication and will be formally accepted for publication once it meets all outstanding technical requirements.

Kind regards,

Katherine A. Borkovich, Ph.D.

Academic Editor

PLOS ONE

Additional Editor Comments (optional):

Reviewers' comments:

Reviewer's Responses to Questions

**Comments to the Author**

1. If the authors have adequately addressed your comments raised in a previous round of review and you feel that this manuscript is now acceptable for publication, you may indicate that here to bypass the “Comments to the Author” section, enter your conflict of interest statement in the “Confidential to Editor” section, and submit your "Accept" recommendation.

Reviewer #1: All comments have been addressed

2. Is the manuscript technically sound, and do the data support the conclusions?

Reviewer #1: Yes

3. Has the statistical analysis been performed appropriately and rigorously? 

Reviewer #1: Yes

4. Have the authors made all data underlying the findings in their manuscript fully available?

Reviewer #1: Yes

5. Is the manuscript presented in an intelligible fashion and written in standard English?

Reviewer #1: Yes

6. Review Comments to the Author

Reviewer #1: The authors have revised the MS of the paper entitled “Genome sequencing and functional characterization of a Dictyopanus pusillus fungal extract offers a promising alternative for lignocellulose pretreatment of oil palm residues” satisfactorily as per suggested comments in my earlier review. The paper can be considered for publication. Also, during the further processing I suggest the author must make sure the format of referencing is in line with the PLosOne.

7. PLOS authors have the option to publish the peer review history of their article (what does this mean?). If published, this will include your full peer review and any attached files.

Reviewer #1: Yes: Pradeep Verma

---

## [Editor Report · Acceptance letter]

20 Jul 2020

PONE-D-19-35214R1 

Genome sequencing and functional characterization of a Dictyopanus pusillus fungal enzymatic extract offers a promising alternative for lignocellulose pretreatment of oil palm residues 

Dear Dr. Doucet:

I'm pleased to inform you that your manuscript has been deemed suitable for publication in PLOS ONE. Congratulations! Your manuscript is now with our production department. 

Kind regards, 

on behalf of

Dr. Katherine A. Borkovich 

Academic Editor

PLOS ONE